# FtbZIP12 Positively Regulates Responses to Osmotic Stress in Tartary Buckwheat

**DOI:** 10.3390/ijms232113072

**Published:** 2022-10-28

**Authors:** Wenfeng Weng, Xiang Lu, Meiliang Zhou, Anjing Gao, Xin Yao, Yong Tang, Weijiao Wu, Chao Ma, Qing Bai, Ruiqi Xiong, Jingjun Ruan

**Affiliations:** 1College of Agronomy, Guizhou University, Guiyang 550025, China; 2Institute of Crop Science, Chinese Academy of Agriculture Science, Beijing 100081, China

**Keywords:** *Fagopyrum tataricum*, *FtbZIP12*, osmotic stress, *Arabidopsis thaliana*, plant resistance

## Abstract

*ABFs* play a key role in regulating plant osmotic stress. However, in Tartary buckwheat, data on the role of *ABF* genes in osmotic stress remain limited and its associated mechanism in osmoregulation remain nebulous. Herein, a novel *ABF* family in Tartary buckwheat, *FtbZIP12*, was cloned and characterized. *FtbZIP12* is a transcriptional activator located in the nucleus; its expression is induced by NaCl, mannitol, and abscisic acid (ABA). Atopic expression of *FtbZIP12* in *Arabidopsis* promoted seed germination, reduced damage to primary roots, and improved the tolerance of seedlings to osmotic stress. The quantitative realtime polymerase chain reaction (RT-qPCR) results showed that the expressions of the typical genes related to stress, the SOS pathway, and the proline synthesis pathway in *Arabidopsis* were significantly (*p* < 0.05) upregulated under osmotic stress. *FtbZIP12* improved the osmotic pressure resistance by reducing the damage caused by reactive oxygen species to plants and maintained plant homeostasis by upregulating the expression of genes related to stress, osmotic regulation, and ion homeostasis. This study identified a key candidate gene for understanding the mechanism underlying osmotic-stress-regulated function in Tartary buckwheat, thereby providing a theoretical basis for improving its yield and quality.

## 1. Introduction

Stress tolerance of plants is influenced by the expression of stress-induced genes regulated by specific transcription factors (TFs) [1]. TFs directly or indirectly regulate the expression of downstream stress-related target genes via cis-acting elements [1,2]. The *bZIP* transcription factor is one of the key regulators of plant stress resistance involved in regulating the defense against pathogens, light tolerance, and stress-signal transduction [3]. The domain of bZIP is conserved in eukaryotes and consists of 6080 amino acids. Its N-terminus contains a N-x7-R/K-x9 motif and a C-terminal leucine zipper region that mediates homologous and/or heterodimerization of the bZIP protein [4,5]. Therefore, mining *bZIP* TFs related to abiotic stress is of great significance for understanding their regulatory mechanisms during stress.

Among the *bZIP* TFs family, the *ABF* subfamily is involved in various pathways such as growth and development and abiotic stress. In *Arabidopsis thaliana*, 13 members of the *ABF* subgroup have been identified, most of which are induced by abiotic stress, such as *AREB1/ABF2*, *AREB2/ABF4*, *ABF1*, and *ABF3* [6,7]. These genes not only improve the resistance of *Arabidopsis* to abiotic stress alone, but also regulate plant resistance in the form of homodimers [7,8]. The regulation of *ABF* subfamily genes in the abscisic acid (ABA) signaling pathway has been extensively studied. *ABF1*–*4* gene expression and accumulation of their endogenous proteins are induced by ABA and play a negative regulatory role in the ABA signal-transduction pathway [9]. During ABA induction, ABF regulates the expression of *SAG29* and other *SAG12*, leading to the senescence and death of plant leaves [10,11]. Among them, the three genes *ABI3–5* are key factors in the ABA signaling pathway involved in seed germination, seedling development, abiotic stress, and flower heading [12,13,14]. In other species, *ABFs* have been reported to significantly increase plant abiotic resistance. For instance, in rice, *OsABF1* significantly increases plant resistance to abiotic stress yet plays a negative regulatory role in the flowering cycle [15,16,17,18]. In *Brassica oleracea*, *BolABI5* expression can be induced by ABA and drought; however, *BolABI5* becomes resistant to drought stress by binding to ABRE cis-acting elements [19]. In cassava, the expression of *MeABL5* was induced by various abiotic and hormonal stresses, which positively regulated *MeCWINV3* and thus improved the multiple resistance of the plant [20]. These findings indicated that the *ABF* subfamily genes are indispensable in plants. Therefore, it is important to study the mechanism underlying *ABF* responses to adversity.

*Fagopyrum tataricum* (Tartary buckwheat) (2n = 8), which belongs to the Polygonaceae family, is a common multigrain crop containing more abiotic active substances and balanced essential nutrients than other food crops [21,22]. Due to its health benefits, the demand for Tartary buckwheat is growing. Tartary buckwheat has the inherent ability to resist adversity, hence it is mostly planted in arid and semi-arid areas [23]. Due to the underdeveloped irrigation conditions in these areas, the limited ability of Tartary buckwheat to withstand stress is far lower than the pressure from the environment. The growth cycle of Tartary buckwheat is affected by abiotic stresses such as water and salt; for example, seed germination is inhibited, photosynthesis is weakened, and the stamen cannot develop normally [24]. Therefore, the yield and quality of Tartary buckwheat are seriously reduced by abiotic stress, hence studying the mechanism underlying Tartary buckwheat’s response to abiotic stress will help to improve the quality and yield of Tartary buckwheat.

In Tartary buckwheat, 96 *bZIP* TFs have been identified and divided into 11 subgroups, among which the *ABF* group is the second-largest subgroup, containing 19 genes [25]. Two TFs, *FtbZIP5* and *FtbZIP83*, have been identified in the *ABF-bZIP* subgroup of Tartary buckwheat, both of which are induced by ABA, NaCl, and drought and are overexpressed in *Arabidopsis,* in which they respond positively to abiotic stress [26,27]. However, the specific characteristics of the other members of this subgroup remain to be determined. Studies on the regulatory role of *FtbZIP12* in plant growth and development will help to elucidate the mechanisms underlying the function of *ABF* TFs during abiotic stresses in Tartary buckwheat. In this study, *FtbZIP12* was isolated from ‘Chuanqiao-2’and a phylogenetic tree was constructed using the *Arabidopsis bZIP* gene family as a reference. Based on the analysis of *ABF* conserved motifs in several species, *FtbZIP12* was preliminarily identified as a member of the *ABF* subgroup. The expression pattern of *FtbZIP12* induced by ABA, mannitol, and NaCl was analyzed using RT-qPCR and transferred into yeast to assess its self-activation characteristics. *FtbZIP12* was also overexpressed in *Arabidopsis* to determine its potential role in improving the stress resistance of *Arabidopsis*. This study identified a new *ABF* subfamily gene in Tartary buckwheat that provides a potential candidate for screening stress-resistance genes and improves both the yield and quality of Tartary buckwheat under abiotic stress.

## 2. Results

### 2.1. Isolation and Characterization of FtbZIP12

To study its functional properties and role in Tartary buckwheat, *FtbZIP12* was cloned and analyzed via homologous cloning. *FtbZIP12* contains a complete open reading frame with a 1299 bp sequence comprising 432 amino acids with a molecular weight of 46,612.39 kDa, a theoretical pl of 9.22, and a formula of C2003H3240N606O643S16 (Expasy ProtParam tool).

A phylogenetic analysis of the Arabidopsis bZIP family revealed that *FtbZIP12* (FtPinG0201993700. 01. T 01) clustered with *ABH04550.1* (unknown), *AEE86309.1* (*ABF3*), *Q9M7Q2.1* (*ABF2*), *Q9M7Q5.1* (*ABF1*), and *Q9LES3.1* (*ABI5*) in the *ABF* subfamily (Appendix A). Among these, *FtbZIP12* was the closest to *ABH04550.1*, although the functional properties of *ABH04550.1* remain to be determined. To better understand the characteristics of *FtbZIP12*, a phylogenetic analysis and a multiple sequence alignment analysis were performed with *ABFs* from other species with known functions (Figure 1A,B). The results showed that the subfamily of AREB/*ABF* TFs with known functions were divided into three subfamilies in different species, among which *FtbZIP12* was closely related to *Arabidopsis*, grapevine, and tobacco (Figure 1A). *FtbZIP12* was most similar to the *ABFs* in Arabidopsis (*NP849777* and *BT026443*), whereas *NP849777* was an osmotic-stress-related TF (Figure 1A). The multiple sequence analysis of *ABF2* revealed that all had the same bZIP conserved domain with four phosphorylation sites (C1–C4), suggesting that *FtbZIP12* was likely have the same functional characteristics (Figure 1B). We subsequently conducted a more detailed analysis of the properties of *FtbZIP12* in Tartary buckwheat to determine its mechanism of action.

### 2.2. FtbZIP12 Expression Is Induced by Abiotic Stress Stimuli

The above bioinformatics analysis suggested that *FtbZIP12* may be related to abiotic stress; therefore, we analyzed the expression pattern of *FtbZIP12* in Tartary buckwheat following NaCl, mannitol, and ABA treatment using RT-qPCR. *FtbZIP12* expression was significantly upregulated by NaCl, mannitol, and ABA in Tartary buckwheat (*p* < 0.05) and its expression trends differed. Under NaCl treatment, with a peak at 3 h, its expression decreased over time (Figure 2A). However, the reverse expression pattern was observed under mannitol treatment, which was more than 10-fold higher at 24 h than 0 h (Figure 2B). Following ABA stimulation, the expression level first increased then decreased, peaking at 3 h (Figure 2C). The stimulated response of *FtbZIP12* to abiotic stress described above showed that the mechanism of *FtbZIP12* in abiotic stress response is relatively complex.

### 2.3. Tissue-Specific Analysis of FtbZIP12

RT-qPCR was used to analyze the tissue specificity in *FtbZIP12* in Tartary buckwheat. The results showed that *FtbZIP12* was expressed in various tissues, although at different levels (Figure 2D). Taking the seed as the control, the relative expression was the highest in old leaves and stems followed by young leaves, and was relatively low in flowers and roots (Figure 2D).

### 2.4. FtbZIP12 Has Transactivation Activity

To further understand the characteristics of *FtbZIP12*, its subcellular localization was predicted using the plant PLOC server (SJTU. Edu. CN). The results showed that *FtbZIP12* was located in the nucleus. To determine whether *FtbZIP12* had self-activation ability, the complete CDS of *FtbZIP12* was inserted into pGBKT7 and pGADT7 vectors and *FtbZIP12*-pGBKT7 + pGADT7/*FtbZIP12*-pGADT7 + pGBKT7 were co-transformed into Y2HGold cells (Saccharomyces cerevisiae). The results showed that the reporter gene could be activated by the two constructs; however, the activation ability of the *FtbZIP12*-pGADT7 construct was weaker than that of *FtbZIP12*-pGBKT7, indicating that *FtbZIP12* was a transcriptional activator (Figure 3).

### 2.5. FtbZIP12 Confers Osmotic Stress Tolerance to Arabidopsis

The overexpression plants of the *FtbZIP12* (35S: *FtbZIP12*) construct were obtained to study their function using gain-of-function methods. Transgenes F1, F2, and F3 were positively screened using 100 mg/L of hygromycin and the overexpressing plants were selected using RT-qPCR. Among them, the T3 generation lines with high (OE#18) and moderate (OE#06) expression levels were selected for subsequent experiments (Appendix A).

To verify the resistance of *FtbZIP12* and determine whether it could improve the stress tolerance of seeds, T3 generation seeds were sown on a half-strength MS solid medium containing 125 mM of NaCl and 225 mM of mannitol; the germination rate was calculated from the second day. As shown in Figure 4, the seed germination rate and growth status of all lines were consistent on the normal medium (Figure 4A,B). Under salt stress, some wild-type seeds did not germinate and the germination rate reached approximately 85% on day 6; the germination rates of transgenic plants (97%) were significantly higher than those of the wild-type plants (Figure 4C,D). Under mannitol-simulated drought conditions, the germination rate of the wild-type plants was only 58%, whereas that of the transgenic lines was >98%, which was significantly higher than that of the wild-type plants (Figure 4E,F). This indicated that *FtbZIP12* may significantly reduce damage to seed germination under osmotic stress.

We next determined whether *FtbZIP12* could improve plant resistance at the seedling stage. The 3-day-old seedlings growing in the normal medium were selected and transplanted to the medium containing 125 mM NaCl and 225 mM mannitol; their root lengths and fresh weights were measured on day 7. As shown in Figure 5A, under normal growth conditions, the roots of all lines reached the bottom of the culture plate. The leaves and stems became larger and longer, whereas the fresh weights and root lengths remained almost unchanged (Figure 5A). Under salt stress, the growth and development of all lines were inhibited, with the wild-type plants showing the most evident damage; their root lengths were only approximately 2.96 cm, which was much lower than that of the transgenic plants (>5 cm) (Figure 5B,D). The fresh weight was only 14.67 mg, which was significantly lower than that of overexpression lines 6 (OE#06) (19.67 mg) and 18 (OE#18) (21 mg) (Figure 5B,D). Under drought stress, the root lengths of transgenic plants (3.45 cm) were also higher than those of the wild-type plants (2.75 cm) (* *p* < 0.05) (Figure 5C,D). The fresh weight of the wild-type plants was only 6.66 mg, whereas those of OE#06 and OE#18 were 15.5 and 21 mg, respectively (** *p* < 0.01) (Figure 5C,E). These results suggested that *FtbZIP12* may reduce the sensitivity of *Arabidopsis* seedlings to osmotic stress, thereby improving plant resistance.

Finally, to assess the physiological and biochemical effects of osmotic stress on the overexpression plant seedlings, soil-cultured seedlings were subjected to NaCl and drought stress and their physiological and biochemical indicators were tested. The results shown in Figure 6A illustrate that under regular watering, the growth vigor and leaf color of the wild-type and transgenic lines did not differ. The growth of all lines was seriously hindered under drought stress, indicating that the plants were short and the leaves were withered. The wild-type plants were the most severely damaged (Figure 6A). Under salt stress, the leaves of the wild-type plants turned yellow or even withered, whereas some new leaves of the overexpressing plants remained green (Figure 6A).

By detecting the content of malondialdehyde (MDA) and the enzymatic activity of proline (PRO) and catalase (CAT), we explored the accumulation of reactive oxygen species (ROS) and the degree of membrane oxidative damage in the transgenic lines under osmotic stress, as shown in Figure 6B–D. Under normal watering conditions, the MDA, CAT, and PRO levels were similar in all plants (*p* > 0.05). After three weeks of NaCl solution application and two weeks of no watering, the OE lines exhibited the highest CAT and PRO contents (Figure 6B,C), whereas the MDA levels were the highest in the wild-type plants and more than three-fold higher than that in the transgenic plants (Figure 6D). These results indicated that under drought and NaCl stress, ROS rapidly increased in the plants. The overexpression of *FtbZIP12* could reduce the damage attributed to ROS by increasing the accumulation of PRO and CAT.

### 2.6. Analysis of the Changes in the Expression Levels of Genes Downstream of FtbZIP12 Related to Stress

To determine the response mechanism of *FtbZIP12* under osmotic stress, we selected genes from *A. thaliana* that may be involved in osmotic stress, including ion transporter genes (*SOS1*, *SOS2*, and *SOS3*), abundant late embryogenesis genes (*LEA6* and *LEA7*), PRO synthetic genes (*P5CS1*), *RD29B*, and *ERD1*. The expression level of these downstream genes under osmotic stress was detected using RT-qPCR. Under osmotic stress, cells lose water, thereby lowering the osmotic pressure. Ions are disordered in vivo; however, *SOS3*, *SOS2*, and *SOS1* in the SOS pathway maintain ion stability [28]. Our results showed that *SOS3*, *SOS2*, and *SOS1* were significantly induced in *FtbZIP12*-overexpressing lines; these genes were increased several-fold to dozens-fold under stress (Figure 7A–C), suggesting that *FtbZIP12* can activate the SOS pathway. After stress treatment, the relative expression of *AtRD29B*, *AtLEA6*, *AtLEA7*, *AtP5CS1*, and *AtERD1* was also induced and the transgenic lines were significantly higher (Figure 7D–H). These results indicated that the enhanced stress tolerance of transgenic lines may be linked to the expression of downstream stress genes induced by *FtbZIP12*.

## 3. Discussion

With higher levels of pollution, the size of saline-alkali soil and arid areas have been increasing annually, which seriously threatens the yield and quality of crops. The *bZIP* TFs in rice, soybean, grape, and other crops have different mechanisms for biological and abiotic stresses [29,30,31,32]. In this study, *FtbZIP12* was identified and isolated. Based on the results of a phylogenetic tree analysis and a multiple sequence analysis, *FtbZIP12* was presumed to be a member of the *ABF* subfamily (Figure 1 and Appendix A). However, the function of *ABH04550.1*, which was closest to *FtbZIP12*, has not been identified, hence the specific functional characteristics of *FtbZIP12* warrant further studies (Appendix A). According to the results of the multiple sequence analysis of the *ABF* gene in each species, *FtbZIP12* was closely related to *A. thaliana*, grape, and tobacco (Figure 1A) with the same conserved domain (Figure 1B). In addition, the subcellular localization prediction results for *FtbZIP12* and its transcriptional activity in yeast were the same as those of Arabidopsis *ABF*, both of which have transcriptional activators located in the nucleus (Figure 2D and Figure 3) [30]. This indicated that *FtbZIP12* is likely to be a member of the *ABFs*. In this study, a high expression of *FtbZIP12* in old leaves and stems of Tartary buckwheat was detected (Figure 2D). We speculated that *FtbZIP12* was related to leaf senescence, stress response, and stress signal transmission. Under abiotic stress, old leaves are usually the first to show stress symptoms, following which a series of signals are transmitted through the stalk to various plant parts in response as part of the defense mechanism. Therefore, whether *FtbZIP12* participates in the reception and transmission of abiotic stimulus signals requires further investigations.

In *Arabidopsis*, the *ABF* subfamily genes *ABF2*, *ABF3,* and *ABF4* were induced under high salinity, drought, freezing, and cold conditions, which improved the ability of the plants to resist external stresses [10,33,34]. Even though *ABF1* was induced at a low expression level, it played a key role in drought stress [35]. The expression of *FtbZIP12* was induced by ABA, NaCl, and mannitol; however, the RT-qPCR results showed that changes in the *FtbZIP12* expression after 0–24 h of mannitol induction were different from those under NaCl stress (Figure 2A,B). This expression pattern was different from that of *FtbZIP83* and *FtbZIP5*, the relative expression of which first increased then decreased under ABA, NaCl, and drought stress [26,27]. We concluded that although *FtbZIP12* belongs to the same subfamily as *FtbZIP83* and *FtbZIP5*, they are not only different in sequence, but also have different roles under osmotic stress. Therefore, we speculated that the regulatory mechanism of *FtbZIP12* in abiotic stress is different from that of *FtbZIP83* and *FtbZIP5*. At present, there is no mature and stable genetic transformation system for Tartary buckwheat, hence it is difficult to obtain stable silencing and overexpression in buckwheat plants. The perfect transformation system of *A. thaliana* provides the possibility of the functional verification of Tartary buckwheat stress-resistance genes. In this study, *FtbZIP12*-overexpressing T3 generation of stably expressing plants was obtained using the floral dip method to verify the regulatory mechanism of *FtbZIP12* in osmotic stress. Previously, several *ABF* functions were verified in *A. thaliana*; *ABFs* can significantly improve plant resistance to external environment. In *A. thaliana*, overexpression of *ABI5* and *ABI3* increased the seed germination rate and reduced post-germination growth retardation via the ABA signaling pathway under osmotic stress [36]. In *Arabidopsis* mutants, the deletion of *AtABF1* increased the sensitivity of primary roots to abiotic stress and seriously affected the expression of dehydration-related genes [34]. Similar to previous results, the ability of *Arabidopsis* with ectopic *FtbZIP12* expression to resist osmotic stress was significantly increased. In terms of the phenotype, the germination rate of seeds was increased, the inhibition of primary root growth was reduced, and seedlings were slightly damaged (Figure 4, Figure 5 and Figure 6).

Under osmotic stress, excessive accumulation of ROS can damage the structure of plant cell membranes and be toxic to plants; the degree of plant damage can be expressed by the MDA content [37]. To resist these poisons, plants produce a series of response mechanisms. CAT is a superoxide that scavenges ROS; increasing its activity helps in such scavenging [38]. PRO is an important osmotic regulator; the increased accumulation of PRO under stress conditions not only helps to regulate intracellular and extracellular osmotic pressure, but also participates in the regulation of different metabolic processes [39]. CAT, MDA, and PRO are key indicators of osmotic stress. Studies have shown that the *ABF* subfamily can participate in plant defense responses to salt stress by regulating ROS, antioxidant enzymes, and PRO [40,41], which is consistent with the results of this study. Here, the transgenic plants were less injured by stress than the wild-type plants, possibly because *FtbZIP12* promoted CAT and PRO accumulation in vivo, reduced MDA production, and decreased cell membrane damage (Figure 6B–D).

In addition, *ABFs* have been shown to improve plant stress tolerance by acting on downstream stress genes and promoting their expression [41]. *RD29B*, *LEA6*, *LEA7*, and *ERD1* are typical stress genes; their promoters contain ABRE, which can effectively bind to *ABFs* under osmotic stress [42,43,44]. Under normal conditions, the expression of these stress genes in transgenic plants did not differ from that in wild-type plants. After 6 h of osmotic stress, all plants had an upward trend, whereas the expression of wild-type plants was significantly lower than that of transgenic plants (Figure 7D–F,H). Transgenic plants may upregulate these stress genes, which may protect cells from dehydration. The SOS pathway plays a crucial role in osmotic stress. Indeed, *SOS3* received the Ca^2+^ ion signal, formed a complex with *SOS2*, and stimulated the downstream *SOS1*, ultimately removing excess Na^+^ from the cell and reducing the harm of ions to cells under osmotic stress [28,45,46,47,48]. In this study, the expression levels of *SOS1*, *SOS2*, and *SOS3* related to osmotic regulation in transgenic lines were significantly higher than those of the wild-type plants after 6 h of osmotic stress (Figure 7A–C,G). Therefore, *FtbZIP12* may also improve the resistance of plants to osmotic stress by regulating the SOS pathway to stabilize the ion balance in plants.

Previous studies have shown that *ABFs* are one of the main participants in *P5CS1*-driven proline synthesis under extreme osmotic stress [44]. However, the role of *ABF* transcription factors (TFs) in regulating proline accumulation and proline-mediated osmotic stress adaptation remains unknown. The transcription of *P5CS1* is a prerequisite for proline accumulation under osmotic stress [49,50]. In this study, although the expression pattern of *P5CS1* under osmotic regulation was the same as that of other genes (*RD29B* and *ERD1*), the proline accumulation pathway of *P5CS1* mediated by *FtbZIP12* was complex. The relative expression of *P5CS1* in overexpression lines was significantly higher than that in wild-type plants after 6 h of osmotic stress; after 3 days, the accumulation of proline was much higher than that of the wild-type plants (Figure 6C and Figure 7A). This suggested that the expression of *P5CS1* was regulated by *FtbZIP12*, thus increasing the accumulation of proline under osmotic stress. There are different views on the pathway of *P5CS1*-driven proline synthesis. Under low permeability treatment, the proline accumulation of ABA-deficient mutant (abi1) plants was less than that of wild-type plants [49,50], whereas the proline accumulation in ABA-deficient mutant (aba2) plants under simulated drought conditions was much higher than that under salt stress [51]. In this study, the transcriptions of *P5CS1* and proline content under salt stress were much higher than those under simulated drought. Interestingly, *FtbZIP12* was also more sensitive to NaCl under the induction of mannitol and NaCl in Tartary buckwheat (Figure 2), possibly because the regulation mechanism of proline varies greatly under different osmotic regulation pathways. *FtbZIP12* may also be involved in the osmoregulatory pathway. It has also been reported that under extreme drought conditions, the proline accumulation of the four mutants (abf1, abf2, abf3, and abf4) was much slower than that of the wild-type plants in the early stage. Although proline accumulation can increase in the late stage, mutant plants have shown evident oxidative and membrane damage [44]. This also showed that proline plays an important role in the repair of plants at the initial stage of osmotic adjustment. Therefore, it is necessary to determine how *FtbZIP12* drives *P5CS1* to regulate proline accumulation and the mechanism underlying proline accumulation under different osmotic conditions.

## 4. Materials and methods

### 4.1. Plant Growth and Treatments

The “Chuanqiao-2” seeds were germinated in clear water and then transplanted into pots with a perlite/soil (2:1 *v*/*v*) mixture. For the *Arabidopsis* culture, all seed lines were first treated with 75% ethanol for 15 min then washed numerous times with absolute ethanol, sown on half-strength MS medium for 2 weeks, and transplanted into small pots with a perlite/soil (2:1 *v*/*v*) mixture. All plants were grown normally in the greenhouse. The greenhouse environment was maintained at 70–75% relative humidity, a 16 h light/8 h dark light cycle, and 22 °C.

### 4.2. Vector Construction and Plant Transformation

Full-length *FtbZIP12* coding sequences (CDS) were isolated from “Chuanqiao-2” and inserted into pCAMBIA1307 (35S: *FtbZIP12*-Myc), pGADT7, and pGBKT7 via homologous recombination. For 35S, *FtbZIP12*-Myc-overexpressing plants were generated using the floral dip method [52]. Positive plant selection using 100 mg/L of hygromycin and the pure T3 generation plants were used in subsequent experiments.

### 4.3. Analysis of the Transactivation of FtbZIP12

The combination of plasmids *FtbZIP12*-pGBKT7 + pGADT7/*FtbZIP12*-pGADT7 + pGBKT7 was co-transferred into yeast AH109; pGADT7 + pGBKT7 was negative. A single yeast colony was selected and cultured overnight in SD-LW liquid medium. The shaken yeast colonies were then diluted to 10^−1^, 10^−2^, and 10^−3^ with 0.9% NaCl and grown on SD-LW and SD-ALWH plates to examine their growth. The plates were examined after 3 days of incubation at 30 °C and 3–6 clones were selected and tested for each experiment. This experiment was repeated three times. SD-LW: -trp and -leu; SD-ALWH: -trp, -leu, -his, and -ade. 

### 4.4. Stress-Induced Treatment of Buckwheat and Arabidopsis

Two-week-old Tartary buckwheat was treated with 300 mM of NaCl and 375 mM of mannitol solution and its tissues were collected at 0, 3, 12, and 48 h following treatment. At the same time, treatment with 100 µM of ABA was conducted and tissues were harvested at 0, 1, 3, 6, and 12 h following treatment; the relative expression of *FtbZIP12* was detected. The treatment of 0 h was used as the control. There were 30 plants for each treatment and five plants for each period. The *Arabidopsis* plants were treated with 300 mM of NaCl and 375 mM of mannitol solution and harvested at 0 and 6 h following treatment.

Each line had nine pots for a total of 27 pots, with six plants in each pot and three pots of each line for each treatment. The wild-type material was treated for 0 h as a control. The expression levels of *AtRD29B*, *AtLEA6*, *AtLEA7*, *AtERD1*, *AtP5CS1*, *AtSOS1*, *AtSOS2*, and *AtSOS3* were investigated using RT-qPCR. The treatment methods were conducted as previously described [2]. Three biological replicates were used to detect changes in gene expression.

### 4.5. Real-Time PCR Analysis

Approximately 0.1 g of Tartary buckwheat or *Arabidopsis* leaves was completely ground into powder in liquid nitrogen and the total RNA was extracted using the RNAprep Pure Plant Plus kit (polysaccharide- and polyphenol-rich) (TIANGEN BIOTECH Co., Ltd., Beijing, China). The integrity of the extracted RNA was checked using 1% agarose gel electrophoresis and the RNA purity and concentration were measured using a spectrophotometer (Beijing Kaiao Technology Development Co., Ltd., Beijing, China). The RNA was reverse transcribed into cDNA using a HiScript^®^ III 1st Strand cDNA Synthesis Kit (+gDNA wiper) (Vazyme Biotech Co., Ltd., Nanjing, China) in a 20 μL reaction system. Primer-BLAST was used to design specific primers for the qPCR with *FtH3* and *Atactin 2* as the internal control gene (Table 1).

The quantitative real-time PCR (RT-qPCR) was performed using Cham Q Universal SYBR qPCR Master Mix (Vazyme Biotech Co., Ltd., Nanjing, China), 1.0 μL of template cDNA, 10.0 μL of 2×SYBR mix, 0.4 μL of each primer, and ddH_2_O to top off the reaction volume to 20 μL. The qPCR procedure was conducted as follows: 95 °C for 30 s followed by 40 cycles of 95 °C for 10 s, 56 °C for 30 s, and 95 °C for 15 s; then 60 °C for 60 s and 95 °C for 15 s (Bio-Rad, Hercules, CA, USA). The relative expression levels of the target genes compared to those of the internal control gene were calculated using the 2^−ΔΔCt^ formula [53]. Three biological replicates and three technical replicates were set up in this experiment.

All procedures were performed in accordance with the manufacturer’s instructions. Primer Premier 5.0 software (Premier Corporation, Vancouver, British Columbia, Canada) was used to design specific primers for the PCR. All primers used for the PCR, vector construction, and RT-qPCR analysis are listed in Appendix A.

### 4.6. Stress Tolerance Assays of Transgenic Arabidopsis

To examine the effect of *FtbZIP12* on the seed germination rate, seeds of all lines were sown on half-strength MS intensity agar plates with or without 125 mM of NaCl and 225 mM of mannitol. The seed sterilization procedure was the same as that given in Section 4.1. Each treatment was conducted in five replicates with 30 seeds per line. The germination numbers were counted daily for 7 days.

To determine the effect of osmotic stress on transgenic seedlings, we used the same training method for plants that grew normally for 3 days on half-strength MS medium. Plants with the same growth vigor were selected and transplanted onto plates with or without125 mM of NaCl and 225 mM of mannitol and cultured upright for 7 days. There were three lines in each treatment and nine plants in total. The experiment was repeated three times. After 7 days, the root length of each plant was measured and the fresh weight of each line (three plants) were weighed.

*A. thaliana* was grown on half-strength MS medium for 2 weeks; we then selected plants with the same growth vigor, transplanted them into the soil, and left them to grow for 2 weeks. Seedlings were treated with 300 mM of NaCl for 3 weeks or no watering for 2 weeks. For plants under the drought treatment, we stopped watering 1 week in advance. Each treatment had six pots, totaling 54 pots, with six plants per pot. The seedlings treated for 10 days were used to measure physiological indicators; the amount of PRO, CAT, and MDA in all lines were detected using Boxbio kits (Beijing Boxbio Science & Technology Co., Ltd., Beijing, China) according to the manufacturer’s instructions. Photos were taken two weeks and three weeks later to record the status.

### 4.7. Bioinformatics Analysis

The physical and chemical parameters of *FtbZIP12* were predicted using the Expasy website (accessed on 25 April 2021 at https://web.expasy.org/protparam). The plant PLOC server (accessed on 6 July 2021 at http://www.csbio.sjtu.edu.cn/bioinf/plant/) was used to predict subcellular localization. We used MEGA 11 for the multiple sequence analysis and alignment as well as the phylogenetic tree analysis [54]. The N-J method (1000 times) was used to construct the phylogenetic tree.

### 4.8. Statistical Analysis

Each treatment had 3–5 independent biological repeats. Each repeat contained 3–6 plants. The experiment was conducted at least three times independently. Data statistics and calculations were conducted using Microsoft Excel (Microsoft, Redmond, WA, USA). To identify significant differences between two sets of data, a Student’s t-test was performed using Excel. The error bars denote ±SDs. Significant differences are denoted by * *p* < 0.05 and ** *p* < 0.01. For multiple comparisons, one-way ANOVA with Duncan’s multiple range test was performed using IBM SPSS Statistics 26.0 software (International Business Machines Corporation, New York, NY, USA). Significant differences between treatments are indicated by different letters. Line charts and column charts were drawn using GraphPad Prism 7.0 software (GraphPad Software, LLC, San Diego, CA, USA).

## 5. Conclusions

In this study, we identified that *FtbZIP12*, an *ABF* gene, was induced and expressed by ABA, NaCl, and mannitol. Heterotopic expression of *FtbZIP12* in *A. thaliana* significantly improved the resistance of plants to osmotic stress due to the following possible reasons: (a) inducing *P5CS1* expression and *P5CS1*-driven proline accumulation; (b) inducing the expression of genes related to the SOS pathway and starting SOS pathway; and (c) directly regulating the expression of stress-related genes (Figure 8). However, further studies are warranted to determine the relationship between *FtbZIP12* and the proline synthesis and SOS pathways. The preliminary results of this study on the function and osmotic regulation mechanism of *FtbZIP12* provided novel insights into the osmotic stress regulation mechanism of Tartary buckwheat and will facilitate further research on *ABFs*.

## Figures and Tables

**Figure 1 ijms-23-13072-f001:**
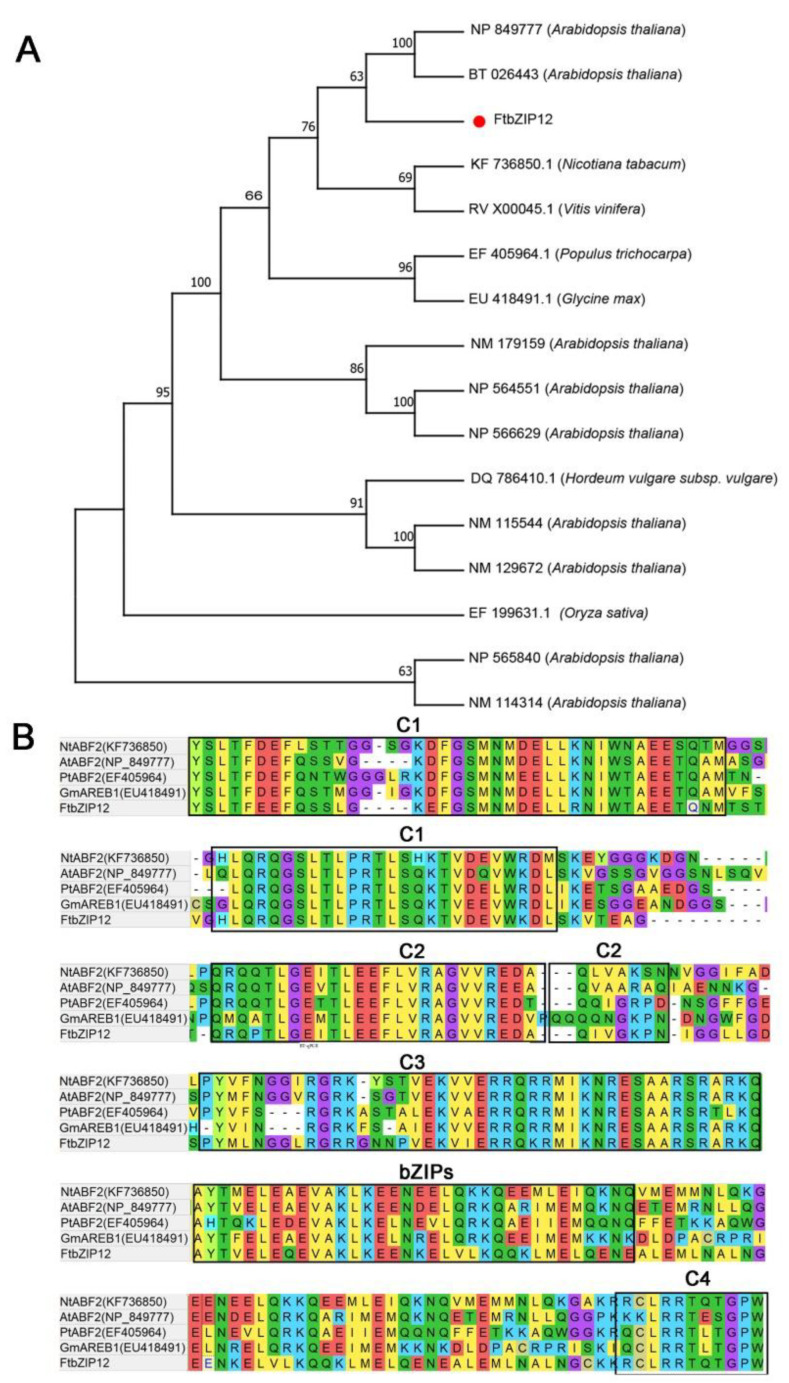
Multiple sequence analysis and phylogenetic tree analysis of *FtbZIP12*. (**A**) Phylogenetic tree analysis of *ABF* subfamily in different species. (**B**) Sequence alignment of *FtbZIP12* protein showing the bZIP conserved domain and the four phosphorylation groups C1–C4 represented by boxes. Among them, the tree-building sequence is cited [25,26].

**Figure 2 ijms-23-13072-f002:**
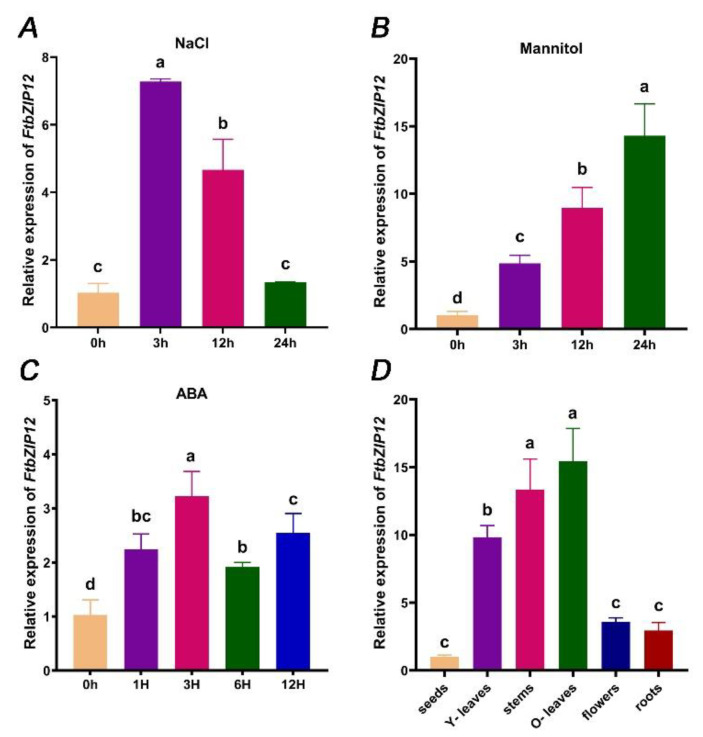
Expression pattern of *FtbZIP12*. (**A**) Expression analysis of *FtbZIP12* in response to 300 mM NaCl, (**B**) 375 mM mannitol, and (**C**) 100 μM ABA while taking the material treated at 0 h as the control. (**D**) The expression levels of *FtbZIP12* in roots, stems, flowers, seeds, and leaves at different developmental stages while taking the seed as the control. Values are expressed as means ± SD (*p* < 0.05); each data value is from three replicate experiments.

**Figure 3 ijms-23-13072-f003:**
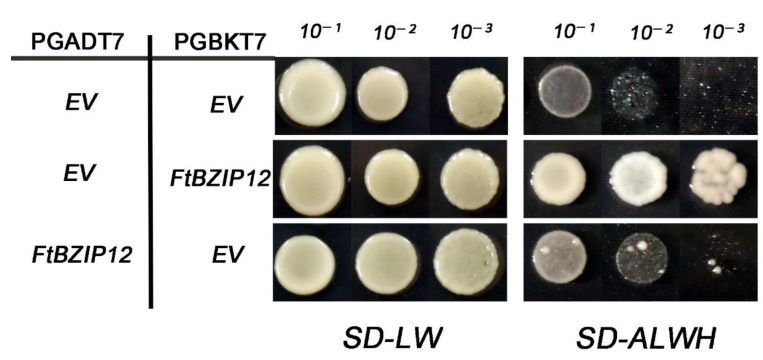
Identification of the transcriptional activation in *FtbZIP12*. Fusion proteins of *FtbZIP12*-pGBKT7 + pGADT7/*FtbZIP12*-pGADT7 + pGBKT7 were co-transformed into Y2HGold cells and grown on SD-LW or SD-ALWH plates to check their growth status. PGADT7 + PGBKT7 was used as the negative control. SD-LW: -trp and -leu; SD-ALWH: -trp, -leu, and -his. The experiment was repeated three times.

**Figure 4 ijms-23-13072-f004:**
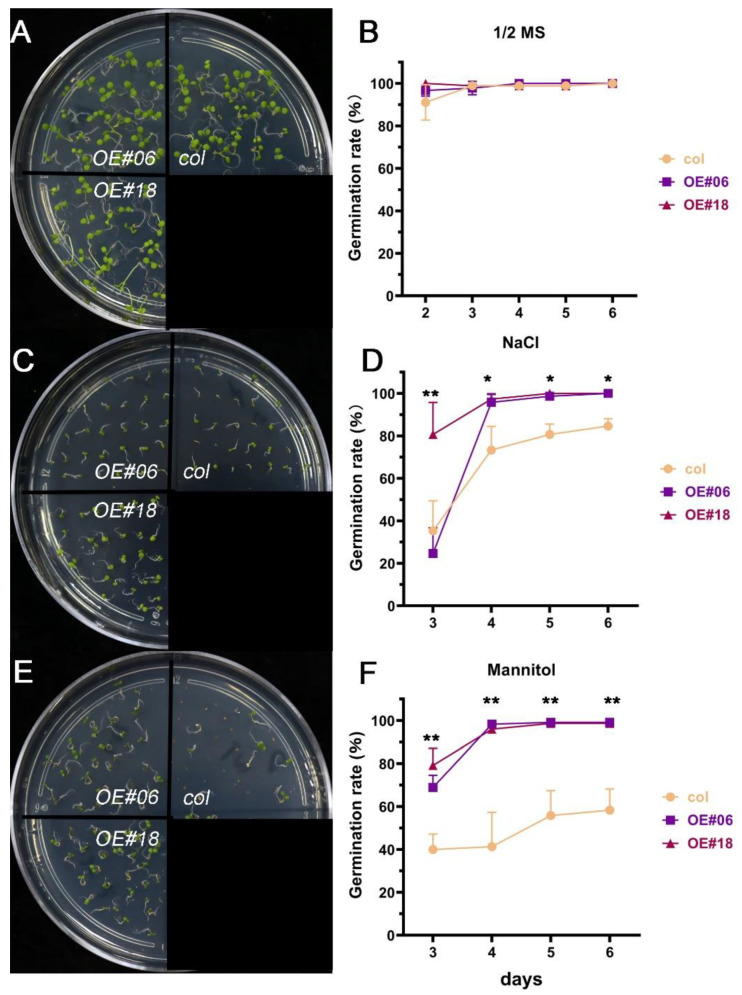
Effects of osmotic stress on *Arabidopsis* seed germination. (**A**) Growth status and germination rate (**B**) of *Arabidopsis thaliana* grown on half-strength MS medium (control). (**C**) Growth status and germination rate (**D**) on half-strength MS medium with 125 mM NaCl. (**E**) Growth status and germination rate (**F**) on half-strength MS medium with 225 mM mannitol. Using a total of 30 seeds per line, the number of germinations was counted from the second day for 7 days. The growth of all lines in half-strength MS medium was used as control. Each value represents three replicate experiments. Values are shown as means ± SD. * *p* < 0.05; ** *p* < 0.01.

**Figure 5 ijms-23-13072-f005:**
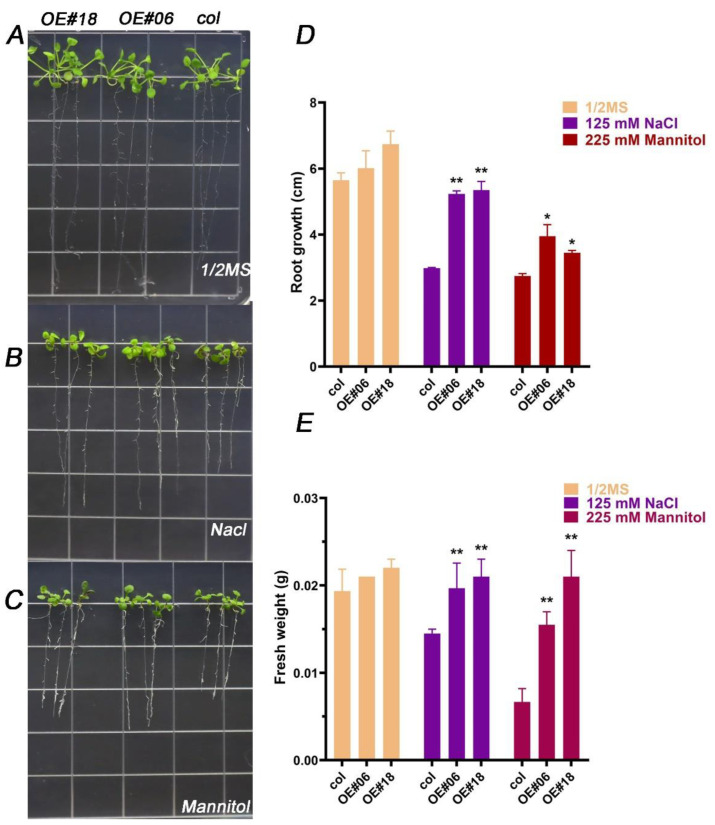
Effects of osmotic stress on the growth of transgenic *Arabidopsis* seedlings under the condition of half-strength MS (**A**), 125 mM NaCl (**B**), and 225 mM mannitol (**C**), as well as the effects of NaCl and mannitol on root length (**D**) and fresh weight (**E**). The growth of seedlings in half-strength MS medium was used as control. Each value represents three replicate experiments. Values are shown as means ± SD. * *p* < 0.05; ** *p* < 0.01 (Student’s *t*-test).

**Figure 6 ijms-23-13072-f006:**
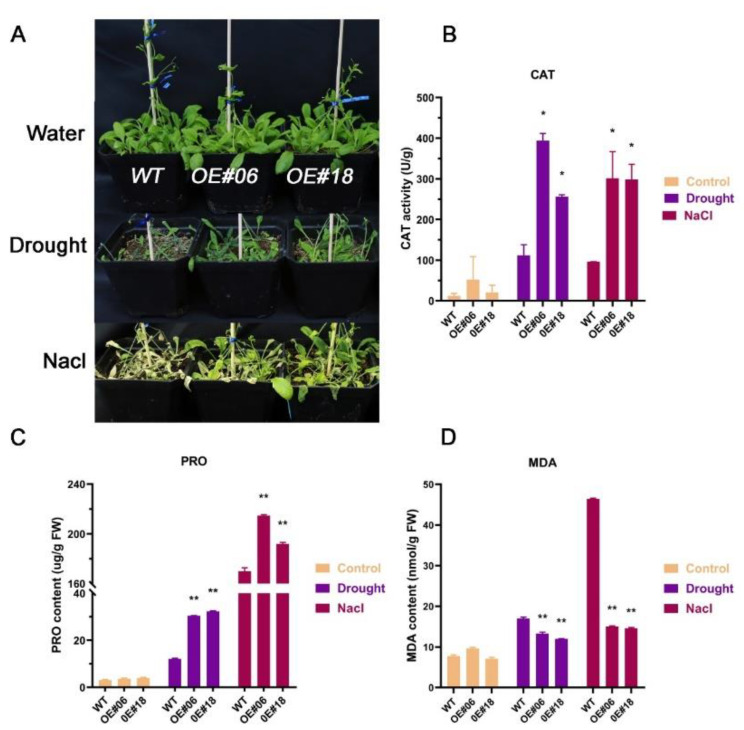
Effects of drought and NaCl on phenotype and physiology of *FtbZIP12*-overexpressing lines. (**A**) Phenotypic map of *Arabidopsis thaliana* under osmotic stress. The contents of CAT (**B**), PRO (**C**), and MDA (**D**) under stress. Each value represents three replicate experiments. Values are shown as means ± SD. * *p* < 0.05; ** *p* < 0.01 (Student’s *t*-test).

**Figure 7 ijms-23-13072-f007:**
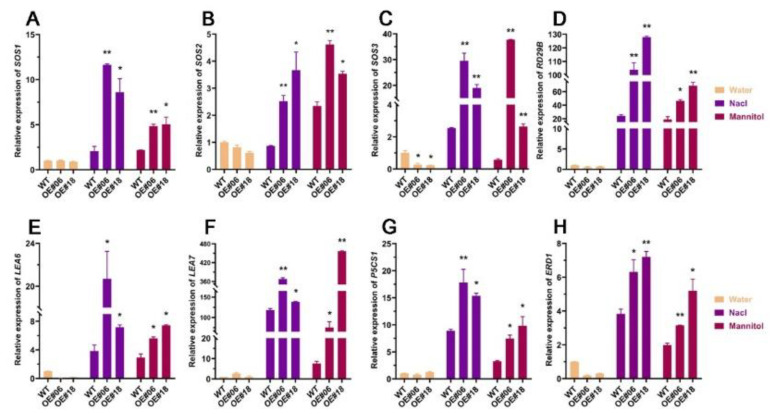
*FtbZIP12* promotes the expression of downstream stress genes. Untreated wild-type plants were considered a control. Three independent experiments were conducted, each of which involved at least three plants. The error bars denote ± SDs; values are shown as means ± SD. * *p* < 0.05; ** *p* < 0.01.

**Figure 8 ijms-23-13072-f008:**
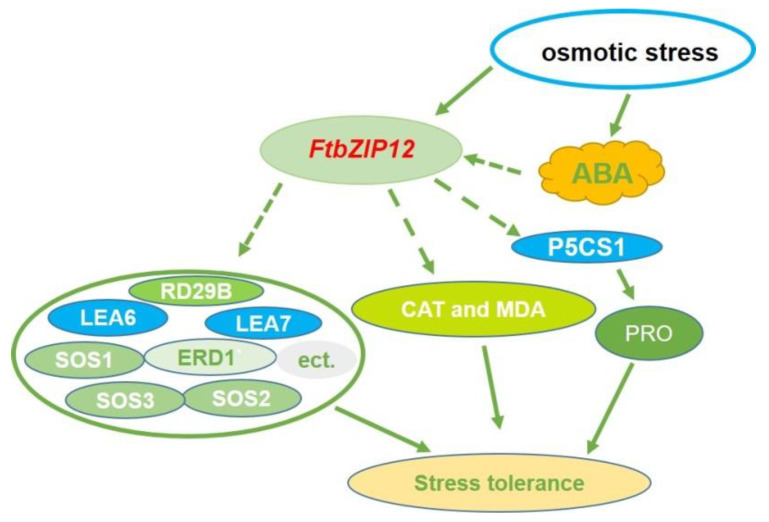
Working model of the mechanism of *FtbZIP12* in improving osmotic stress tolerances. *FtbZIP12* may improve Tartary buckwheat stress resistance by regulating related downstream stress genes, the SOS pathway gene, and the *P5CS1*-driven proline synthesis pathway. Solid arrows indicate proven pathways whereas the dashed arrows show speculative pathways that remain to be confirmed.

**Table 1 ijms-23-13072-t001:** List of primers in Arabidopsis and Tartary buckwheat.

Primer Name	Forward Primer (5′-3′)	Reverse Primer (5′-3′)	Functions
*F3H*	GTATAGCCCCCTAAATGCTCG	GGATTGTGAGATTGTTGCCTATCG	Actin
*AtActin2*	GGAAAGGATCTGTACGGTAAC	TGTGAACGATTCCTGGAC	
*AtRD29B*	GAAACCAAAGATGAGTCGACAC	TTTTTCGTAAACCGGAGTCAAC	Stress-related genes
*AtLEA6*	GCAGAAGCGAATATGGATATGC	CGGAGGATAAGTCGGATGATAG	
*AtLEA7*	TCAAGAGTCCAAAGACAAGACA	GTATATTCAGCTGCATCGTGTG	
*AtERD1*	CTTTCTCTATCAGCACGAAACG	CGGTGCGATATATTGACAATCC	
*AtP5CS1*	AGCTTGATGACGTTATCGATCT	AGATTCCATCAGCATGACCTAG	
*AtSOS1*	ATTTTGATGCAGTCAGTGGATG	GCAAGCAGATTCTAGTCTTTCG	
*AtSOS2*	GCGAACTCAATGGGTTTTAAGT	CTTACGTCTACCATGAAAAGCG	
*AtSOS3*	CCGGTCCATGAAAAAGTCAAAT	CTCTTTCAATTCTTCTCGCTCG	
*FtbZIP12*	TGCCTCGAACACTAAGCCAG	ATGGGGACTAATATGAACTTCAAATC	qPCR
*FtbZIP12*	ATGGGGACTAATATGAACTTCAAATC	TTACCATGGACCCGTCTGT	CDS
*FtbZIP12*	gacttgaactcggtatctagaATGGGGACTAATATGAACTTCAAATC	cttgatatcgaattcctgcagTTACCATGGACCCGTCTGT	Homologous recombination

## Data Availability

The data are present within the article or the Appendix A.

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
