# Peer review of "FtbZIP12 Positively Regulates Responses to Osmotic Stress in Tartary Buckwheat"

_ijms, 2022, doi:10.3390/ijms232113072_

Round 1

Reviewer 1 Report

Comments and Suggestions for Authors

To,

The Editor,

IJMS, MDPI,

Manuscript ID: ijms-1984184

 Subject: Submission of comments of the manuscript in “IJMS"

 Dear Editor IJMS, MDPI,

 Thank you very much for the invitation to consider a potential reviewer for the manuscript (ID: ijms-1984184). My comments responses are furnished below as per each reviewer’s comments. 

In the reviewed manuscript, the authors functionally characterized the FtbZIP12, an ABF subfamily gene from Tartary buckwheat. FtbZIP12 was induced by abscisic acid, NaCl, and mannitol, but its expression patterns were different with each. Further, the yeast two-hybrid experiment showed that FtbZIP12 had trans-activation activity. Overexpression of FtbZIP12 improved plant stress tolerance in Arabidopsis thaliana by regulating the downstream stress-responsive genes. Transgenic plants had an increase in proline content and catalase activity and a decrease in malondialdehyde following stress. In general, as a transcriptional activator, FtbZIP12 is induced by abiotic stress, and by increasing the transcription levels of downstream genes, it initiates relative physiological processes and improves plant stress tolerance. In general, the manuscript represents a very big piece of information in a logical presentation. Therefore, it might be conditionally accepted subject to major revision. Authors have to improve their manuscripts with many non-clear meanings, inaccuracies and inconsistencies, and the authors need to address the following issues before it can be accepted for publication. 

1.    I have read the entire manuscript and my initial comment is that manuscript is poorly written. I have significant concerns about the grammar and vocabulary of the manuscript; therefore, improvement of the language is highly needed. 

2.    The structure of the abstract should be improved, as well as the lack of several aspects that should be included in this section. Most of the abstracts contain confusing and uninformative sentences. Please give more precise objectives here (such as in the Abstract). 

3.    Introduction grammatical issues appear to be most prevalent in the introduction, making for very confusing reading. Further, the introduction is short but has no clear thread. 

4.    General note: the figures in this section are quite low resolution and difficult to make out. Higher-resolution versions will be needed for publication, for example, in Figures 1A, 1B, Figure 2, and Figure S2.

5. Discussion- many times references are made to the information given in the Introduction section (sometimes more general information). It would be good to discuss especially the results and critically, ie. Which can cause differences in the results of authors and other articles.

6. I would like the Authors to provide the methodology and results of the number of replications wherever possible. The same applies to the statistical significance of the results. Please describe statistical methods used in the work in materials and methods.

7. qRT-PCR methodology provided is also very vague and confusing. Please provide more details like what the calibrator used in the study. I assume the authors have used the control as the calibrator. If so, the authors should not include the control within the bar graph as it represents the fold change between the treated vs control and a fold change of “1” for the ‘control’ doesn’t make any sense.  Also, would be good to provide details on what reagents (details of probes used, if any, if SYBR was used then details for that, etc.) and real-time PCR machines were used in the current study.

8.     References - it is necessary to correct the names of the journals and respect the capitalization of their letters. Authors not follow the journal format, hence, check carefully the Journal format and references.

9.    L-444 Vitis vinifera replaced with Vitis vinifera.

10.     L-455 Brassica napus replaced with Brassica napus.

Author Response

Thank you very much for your valuable comments. We have revised the manuscript based on your comments and suggestions. This paper has been revised in review mode and reminded by annotations.

  1. I have read the entire manuscript and my initial comment is that manuscript is poorly written. I have significant concerns about the grammar and vocabulary of the manuscript; therefore, improvement of the language is highly needed.

Response: Thank you very much for your advice. We have consulted relevant experts to revise and polish the grammar of this paper, so as to avoid the occurrence of non-standard language.

  1. The structure of the abstract should be improved, as well as the lack of several aspects that should be included in this section. Most of the abstracts contain confusing and uninformative sentences. Please give more precise objectives here (such as in the Abstract).

Response: Thank you very much for your advice. In this paper, the full text has been reorganized, and the abstract part has been completely rewritten in line 28-44.

  1. Introduction grammatical issues appear to be most prevalent in the introduction, making for very confusing reading. Further, the introduction is short but has no clear thread.

Response: Thank you very much for your advice. The introduction is short, illogical and unclear, which has been revised in the introduction (line 48-114). And has been commented.

  1. General note: the figures in this section are quite low resolution and difficult to make out. Higher-resolution versions will be needed for publication, for example, in Figures 1A, 1B, Figure 2, and Figure S2.

Response:  Thank you very much for your advice. Figure 1, Figure 2 and Figure S2, which were not clear before, have been replaced according to the diagram requirements of the article. The detailed location is line192, line219 and 250 in the text, and they were annotated in the relevant position.

  1. Discussion- many times references are made to the information given in the Introduction section (sometimes more general information). It would be good to discuss especially the results and critically, i.e. which can cause differences in the results of authors and other articles?

Response:  Thank you very much for your advice. This article has explained and discussed the key results of this article. See the discussion for details(lin349-503). This part has been annotated.

  1. I would like the Authors to provide the methodology and results of the number of replications wherever possible. The same applies to the statistical significance of the results. Please describe statistical methods used in the work in materials and methods.

Response:  Thank you very much for your advice. The paper has been supplemented with more detailed descriptions of each experimental procedure in the material and methods section; statistical methods are also described at 4.7 Statistical analysis(line 624).

  1. (1) qRT-PCR methodology provided is also very vague and confusing. Please provide more details like what the calibrator used in the study. I assume the authors have used the control as the calibrator. If so, the authors should not include the control within the bar graph as it represents the fold change between the treated vs control and a fold change of “1” for the ‘control’ doesn’t make any sense. (2) Also, would be good to provide details on what reagents (details of probes used, if any, if SYBR was used then details for that, etc.) and real-time PCR machines were used in the current study.

Response: (1) For the tissue specificity experiments of FtbZIP12, the seeds were used as controls. For the rest of the RT-qPCR experiments, the wild-type material was treated for 0 h as a control. They are also illustrated in the corresponding figure legend. This article has been revised and annotated in line 542-553.

(2) I elucidate the reagents used in qPCR experiment, the details of real-time PCR machine and the detailed experimental steps in lines 556-578. Details are as follows:

Roughly 0.1 g tartary buckwheat or Arabidopsis leaves were completely ground into powder in liquid nitrogen, and total RNA was extracted using the RNAprep Pure Plant Plus kit (polysaccharides and polyphenols-rich) (TIANGEN BIOTECH Co., LTD, Beijing, China). The integrity of extracted RNA was checked using 1 % agarose gel electrophoresis, and RNA purity and concentration was measured using a spectro-photometer (Beijing Kaiao Technology Development Co., Ltd., Beijing, China). RNA was reverse transcribed into cDNA using HiScript ® III 1st Strand cDNA Synthesis Kit (+gDNA wiper) (Vazyme Biotech Co., Ltd, Nanjing, China) in a 20 μL reaction system. Primer-BLAST was used to design specific primers for qPCR, with FtH3 and Atactin 2 as the internal control gene (Table 1).

Quantitative real-time PCR (RT-qPCR) was performed using Cham Q Universal SYBR qPCR Master Mix (Vazyme Biotech Co., Ltd, Nanjing, China), 1.0 μL template cDNA, 10.0 μL 2×SYBR mix, 0.4 μL of each primer, and the reaction volume was made up to 20 μL with ddH2O. qPCR procedure was conducted as the follows: 95℃ for 30 s; followed by 40 cycles of 95℃ for 10 s, 56℃ for 30 s, and 95℃ for 15 s; and 60℃ for 60s; and 95℃ for 15s (Bio-Rad, Hercules, California, USA).The relative expression levels of target genes compared to the internal control gene were calculated using the 2−ΔΔCt formula. Three biological replicates and three technical replicates were set up in this experiment. All procedures were performed in accordance with the manufacturer's instructions.

  1. References - it is necessary to correct the names of the journals and respect the capitalization of their letters. Authors not follow the journal format, hence; check carefully the Journal format and references.

Response:  Thank you very much for your advice.  I have carefully checked the Journal format and corrected the format of references as required(line668-815).

  1. L-444 Vitis vinifera replaced with Vitis vinifera.

Response:  Thank you very much for your advice.   " Vitis vinifera " in the manuscript have all been modified to italics (line-761).

  1. L-455 Brassica napus replaced with Brassica napus.

Response:  Thank you very much for your advice. " Brassica napus " in the manuscript have all been modified to italics (line-783).

Reviewer 2 Report

Comments and Suggestions for Authors

The submitted manuscript to IJMS investigates the positive inlvolvment of FtbZIP12 in adaptation responses of Tartary buckwheat against salt and drought stresses. Although the topic of manuscript is interesting and under the scope of this journal, however, before publication, the following concerns should be resolved:

Line 13-17: should be re-written. Authors are failed to provide the research gap effeciently.

The authors should provide breif introduction of experimental design/system.

In the abstract section, the results explanation could be improved, too.

Figure S1 and 1 are not of good quality.

The name of the gene should be provided with y-axis and this should be followed throughout the manuscript.

Figure 4: the „G” of germination rate should be capital and there should be a gap between rate and bracket at the end of y-axis. This is general understanding and it should be rechecked throughout the manuscript.

The figures quality is not up to the standards. The authors are strongly recomended to prepare the good quality figures including the 8th one.

Line 355-360: Very confusing statements: line 355: irrigated? why the authors used ABA application? What was the rational behind it? What was the osmotic pressure after applying NaCl and mannitol?

Since both the compounds induce osmotic stress to the plant tissues, so it is suggested to use osmotic stress instead of salt and drought stress. The salt and drought stresses should be replaced by osmotic stresses throughout the manuscript including title.

The conclusion section is very poorly written. It should be extensively improved.

Author Response

The submitted manuscript to IJMS investigates the positive involvement of FtbZIP12 in adaptation responses of Tartary buckwheat against salt and drought stresses. Although the topic of manuscript is interesting and under the scope of this journal, however, before publication, the following concerns should be resolved:

1.Line 13-17: should be re-written. Authors are failed to provide the research gap effeciently. The authors should provide brief introduction of experimental design/system. In the abstract section, the results explanation could be improved, too.

Response:  Thank you very much for your advice. This paper has been reworked for the abstract, which is in lines 28-44 of the text.

2、Figure S1 and 1 are not of good quality.

 Response:  Thank you very much for your advice. Figures S1 and 1 have been redone in this paper. See the note in the text (Line 188 and line 192).

3、The name of the gene should be provided with y-axis and this should be followed throughout the manuscript.

Response:  Thank you very much for your advice. In this paper, the name of the gene have be provided with y-axis in Figure 2, Figure S2 and Figure7(line219, line250 and line344).

4、Figure 4: the „G” of germination rate should be capital and there should be a gap between rate and bracket at the end of y-axis. This is general understanding and it should be rechecked throughout the manuscript.

Response:  Thank you very much for your advice. Figure 4, Figure 5, and Figure 6 have been modified accordingly (line 266, line 293 and line 321).

5、The figures quality is not up to the standards. The authors are strongly recommended to prepare the good quality figures including the 8th one.

Response:  Thank you very much for your advice. All figures in the text have been re-uploaded and Figure 8 has been redrawn. In line188, line192, line219, line250, line266, line293, line321, line344 and line510.

6、Line 355-360: Very confusing statements: line 355: irrigated? why the authors used ABA application? What was the rational behind it? What was the osmotic pressure after applying NaCl and mannitol?

Response: Thank you very much for your advice. (1) The word " irrigated " in the manuscript has been changed to “watered” in line 542.

(2) FtbZIP12 belongs to the “A” subfamily of the basic leucine zipper (bZIP) transcription factor families; and is highly homologous with ABF2 and ABF1 of Arabidopsis thaliana. Studies have shown that ABF2 was reported to be involved in abiotic stress responses by regulating the expressions of stress-regulated genes, and ABF is a key transcription factor that mediates ABA signal transduction and ABA regulates drought response (Kim et al., 2010; Yoshida et al., 2015; Wang et al., 2018). ABA significantly induced the expression of ABF gene and the accumulation of endogenous ABF protein (Wang et al., 2018).

   Abscisic acid (ABA) is a key phytohormone involved in a host of biological processes, including plant development and responses to biotic and abiotic stresses (Finkelstein et al. 2002;

Raghavendra et al. 2010). Endogenous ABA levels in plant cells are increased in response to osmotic stresses such as drought and high salinity, leading to expression of stress-responsive genes. Indeed, exogenous application of ABA can induce many dehydration-responsive genes (Zhu 2002; Yamaguchi-Shinozaki & Shinozaki 2006). ABA-dependent gene expression plays an essential part of transcriptional regulatory networks under osmotic stress conditions as well as ABA-independent gene expression (Yamaguchi-Shinozaki & Shinozaki 2006). Promoter analyses of ABA-inducible genes identified the ABA-responsive element (ABRE; PyACGTGG/TC) as a conserved cis-element (Guiltinan et al. 1990; Mundy et al.1990; Busk & Pagès 1998). In this paper, by predicting the up 2000 bp promoter sequence of FtbZIP12, five ABRE cis-elements are found, as shown below; therefore, ABA is also used in this paper.

Kim, S., Kang, J. Y., Cho, D. I., Park, J. H., and Kim, S. Y. (2010). ABF2, an ABRE-binding bzip factor, is an essential component of glucose signaling and its overexpression affects multiple stress tolerance. Plant J. 40, 75–87.

Wang X, Guo C, Peng J, Li C, Wan F, Zhang S, Zhou Y, Yan Y, Qi L, Sun K, Yang S, Gong Z, Li J. ABRE-BINDING FACTORS play a role in the feedback regulation of ABA signaling by mediating rapid ABA induction of ABA co-receptor genes. New Phytol. 2019 Jan;221(1):341-355. doi: 10.1111/nph.15345.

Yoshida T, Fujita Y, Maruyama K, Mogami J, Todaka D, Shinozaki K, Yamaguchi-Shinozaki K. 2015. Four Arabidopsis AREB/ABF transcription factors function predominantly in gene expression downstream of SnRK2 kinases in abscisic acid signalling in response to osmotic stress. Plant, Cell &Environment 38: 3 5–49.

(3) The abiotic stress method of the materials in this paper refers to He Zihang, and does not take into account the change of osmotic pressure, so there is no relevant experimental data.

He, Z.; Wang, Z.; Nie, X.; Qu, M.; Zhao, H.; Ji, X.; Wang, Y. UNFERTILIZED EMBRYO SAC 12 phosphorylation plays a crucial role in conferring salt tolerance. Plant Physiol. 2022, 188(2), 1385-1401.

7、Since both the compounds induce osmotic stress to the plant tissues, so it is suggested to use osmotic stress instead of salt and drought stress. The salt and drought stresses should be replaced by osmotic stresses throughout the manuscript including title.

Response: The salt and drought stresses should be replaced by osmotic stresses, in line1, line 45, line 242, line 266, line 311, line 223 and line 329, line 327 of text.

8、The conclusion section is very poorly written. It should be extensively improved.

 Response: Thank you very much for your advice. The conclusion has been revised(line 649-658).

Round 2

Reviewer 1 Report

Comments and Suggestions for Authors

Dear Editor,

Thank you for providing the opportunity to review the revised manuscript. The manuscript is improved considerably after revision according to the reviewer's comment. Now this study is a suitable contribution to the IJMS. I recommend the manuscript for publication.

Thank you

With best regards